# Understanding financial hardship in families of people living with dementia: Protocol for a scoping review to identify subjective self-report measures that evaluate financial hardship

Noelle E. Carlozzi[1,2*], Christopher M. Graves[1], Jacqueline L. Freeman[3], Amanda N. Leggett[4], Jennifer A. Miner[1], Jin-Shei Lai[5], Madison Fansher[1], Minal R. Patel[6]

**1** Department of Physical Medicine and Rehabilitation, University of Michigan, Ann Arbor, Michigan, United States of America, **2** Department of Surgery, University of Michigan, Ann Arbor, Michigan, United States of America, **3** Taubman Health Sciences Library, University of Michigan, Ann Arbor, Michigan, United States of America, **4** Wayne State University, Detroit, Michigan, United States of America, **5** Department of Medical Social Sciences, Feinberg School of Medicine, Northwestern University, Chicago, Illinois, United States of America, **6** School of Public Health, University of Michigan, Ann Arbor, Michigan, United States of America

* carlozzi@med.umich.edu

## Abstract

### Background

Financial hardship (including financial stress, financial strain, asset depletion, and financial toxicity) is a highly relevant construct among the 6.9 million people living with Alzheimer's disease and related dementias (ADRD) in the United States and their family networks. This scoping review will identify existing measures and approaches for capturing financial strain among these families.

### Methods and Analysis

This scoping review will be reported in accordance with PRISMA-ScR guidelines. Searches will be conducted in Embase (Embase.com), Medline (Ovid), CINAHLComplete (EbscoHost), AgeLine (EbscoHost), APA PsycInfo (EbscoHost), Scopus, and Web of Science Core Collection databases to identify tools, measures, or approaches for measuring self-reported financial hardship among family members of people living with ADRD. Data elements from both the National Health and Aging Trends Study and the associated National Study of Caregiving will also be considered. Two independent raters will screen the search results and discrepancies with be resolved by a third rater. Extraction will include the use of artificial intelligence-based software and verification by an independent human rater; any changes that are made to the AI-generated extraction will be reviewed by a second independent rater. A frequency analysis will be conducted to analyze the data, and a summary of the psychometric properties of each of the identified measures will be presented.

**Data availability statement:** No datasets were generated or analysed during the current study. All relevant data from this study will be made available upon study completion.

**Funding:** This manuscript was supported by grant number grant R01AG088345(PI: Carlozzi, N.E.) from the National Institutes of Health (NIH), National Institute of Aging. The funders had no role in study design, data collection and analysis, decision to publish, or preparation of the manuscript.

**Competing interests:** The authors have declared that no competing interests exist.

## Ethics and Dissemination

The findings from this review will be disseminated through peer-reviewed publications and conference presentations. Ultimately, this work will inform the development of a new patient-reported outcome measurement system designed to provide a comprehensive assessment of the different aspects of financial hardship for families of people living with ADRD.

## Scoping Review Registration

In accordance with the Preferred Reporting Items for Systematic Reviews and Meta-Analyses Protocols (PRISMA-P), the following protocol was registered via https://osf.io on March 14, 2025 (registration DOI: https://doi.org/10.17605/OSF.IO/J26KT).

## Introduction

Financial hardship is a highly relevant factor among the 6.9 million people living with Alzheimer's disease and related dementias (ADRD) and their family networks [1–4]. Financial hardship is multifaceted [5,6], encompassing more than the economic burden associated with ADRD; it also includes financial stress (anxiety or psychological distress regarding expenses), financial strain (objective lack of money to pay for needs [1,7]), asset depletion (cost-associated impact on a person's life savings), and financial toxicity (the impact of out-of-pocket care expenses on quality of life [8,9]). Unfortunately, there are currently no special programs for care coverage for this vulnerable population unlike other chronic disease conditions, such as Medicare's End-Stage Renal Disease Program [10].

While other diseases, such as cancer, have well-established (but highly specific) measures of financial hardship, no such measures exist that are specific to people living with ADRD and their family networks (i.e., immediate family members, biological and nonbiological relatives). Existing measures for cancer such as the Economic Strain Model [11] and the COST-FACIT [12] focus solely on the impact that having cancer has on individual and/or household finances (i.e., financial toxicity), while neglecting other aspects of financial hardship such as financial stress or asset depletion. Such measures also neglect to capture the aspects of financial hardship that are specific to ADRD families, including the intergenerational and/or cross-household financial costs associated with care.

Caring for an individual living with ADRD is estimated to cost $25,000 to $31,000 annually for formal care, and $47,000 to $59,000 for informal care [13,14]. These estimates are typically confined to the costs of in-home or nursing home care (formal costs) and lost wages (for informal costs); as such, the actual cost of caring for an individual living with ADRD may be significantly underestimated in the current literature. Informal caregivers and families of individuals living with ADRD incur significant care costs. For example, ADRD families are estimated to incur 70–90% of total care

costs [15–17], which is significantly higher than other chronic illnesses (i.e., 18–33% of total care costs for families of cancer patients [18]).

The intergenerational and cross-household costs of dementia also likely reflect the changing demographics of families of people living with ADRD. For example, a recent secondary analysis of four UK caregiver studies found that dementia caregivers were on average younger, and more likely to be the son or daughter of their care recipient than caregivers of people with either cancer or acquired brain injury [19]. This shift in demographic composition for primary caregivers will likely exacerbate "hidden" costs such as work/productivity impairment, reduction or loss of salary, and increased time off work, given that a younger demographic is more likely to be an active part of the workforce. Families of people living with ADRD must also bear the cost of additional hospitalizations for reasons other than the disease itself – with some reports finding that over a quarter of hospitalizations of people with ADRD are due to syncope, falls, and trauma [20]. Finally, families of people living with ADRD must manage financial challenges unique to the context of ADRD, such as impulsive spending or poor spending decisions on the part of their care recipients [21].

With a rapidly aging population, the number of people living with ADRD is expected to reach 11.2 million individuals by 2040 [22], thus it is critical that we are able to better support this population and their caregivers. ADRD has a longer clinical course than many other neurodegenerative conditions, thus cost of treatment may be especially burdensome for this population. Indeed, a recent study demonstrated that the net worth of individuals with ADRD decreased an average of 60% in the first eight years of diagnosis, and that their out-of-pocket medical expenses nearly doubled (1.9x) when compared to a demographically matched control group [23]. Given this longer clinical course, substantial informal care cost, and disease-specific financial challenges in which dementia caregivers must contend with, it is important to have measures of financial hardship that are related to the unique circumstances for families of people living with ADRD. Absent valid, disease-specific measures, it is difficult to identify the most important aspects of financial hardship for individuals and families impacted by ADRD. Better measurement is therefore critical to informing the policies and programs needed to protect these families from financial hardship, which may ultimately improve the health and well-being of patients and their families.

To address this gap, we are developing a new patient-reported outcome (PRO) measurement system to capture and characterize the most important aspects of financial hardship related to ADRD. This measurement system will include developing new, sophisticated computer adaptive tests (i.e., smart tests) that can be used to capture these financial constructs. As one of the first steps in this process, we are conducting a scoping review to identify existing PROs that have previously been used in the dementia care literature to assess self-reported financial hardship. After identifying these existing measures, we will also review the psychometric properties of these measures, including whether or not the measures are specific to families of people with dementia.

## Methods

The protocol from this scoping review follows the PRISMA-P reporting guidance [24–26] and was registered with OSF Registries on March 14, 2025 (registration DOI: https://doi.org/10.17605/OSF.IO/J26KT). Please see the PRISMA-P checklist as supporting information (S4 PRISMA-P Checklist). We outline the specific steps for this process below.

This scoping review will be conducted in accordance with established PRIMSA-ScR guidelines [2,3]. Any amendments made to the original protocol will be added to the registration on OSF Registries, and will be acknowledged in the scoping review.

### Search methods

**Identification.** A health science library information scientist, in collaboration with members of the study team, will conduct searches in Embase, Ovid Medline, CINAHL, AgeLine, PsycInfo, Scopus, and Web of Science databases to identify self-report measures of financial hardship that have been used in dementia research. Data elements from both

the National Health and Aging Trends Study and the National Study of Caregiving will be considered. Search terms will include derivates and controlled vocabulary for select dementia-related clinical diagnoses, financial hardship, and self-report measures of financial hardship or caregiver outcomes. An example search is provided in S1 Appendix.

**Eligibility.** To be included in our search, articles must focus on ADRD and include subjective data on financial factors reported by caregivers. For example, we will not consider studies where data are collected from clinicians, and we will not consider articles that solely focus on hours of care (converted to monetary value), objective financial data (such as costs), nor articles that purely examine economic data (such as Medicare claims).

Articles must be peer-reviewed and have full text available. Although there will be no limits set on publication date, we will exclude articles found in the grey literature, as well as reviews, case studies, response letters, research reports, graduate theses, and study protocol papers. We will not exclude articles based on geographical location assuming that they are available in English.

**Selection criteria.** A review management software (Covidence) will be used to aid in search result review. Title and abstract review will include two independent raters. Following title and abstract screening, full-text screening will be conducted (again with two independent reviewers). All inclusion/exclusion discrepancies found in either the title and abstract screening or full-text review will be reviewed and reconciled by two raters, with a third, independent rater in cases where consensus is not met. Screening is anticipated to be completed by May 2025.

## Data extraction

Data extraction will involve the use of a university-specific artificial intelligence software. Extraction will include author details; year of publication; sample size; subjective, self-report measures of financial constructs; and sample-specific (if available) reliability (i.e., internal consistency or test-retest reliability) and validity data (convergent or discriminant validity, known-groups validity, or responsiveness data) for the financial construct (See S2 Appendix for a draft of the AI-specific queries for extraction). All AI-generated extraction will be reviewed by an independent human rater, and any corrections made to the AI-based extraction output will be reviewed by a second, independent rater. A data extraction summary sheet (Microsoft Excel) will be used to compile the data. Data extraction is anticipated to be completed by June 2025.

## Data analysis

The data analysis will focus on the following three domains: [1] existing measurements of financial hardship [2], the reliability/validity of the measures, and [3] whether the measurement is specific to caregivers of individuals living with ADRD.

We will calculate the frequency of the existing measures of financial hardship, and also summarize the frequency in which various facets of financial hardship are assessed by these measures: financial strain, financial stress, asset depletion, and financial toxicity. We will also categorize measures as capturing objective (e.g., loss of income) or subjective (e.g., stress/distress) data, and whether the measures are specific to caregivers of individuals with ADRD. When data are available, we will assess the psychometric properties of existing measures and will categorize them based on their demonstration of reliability (i.e., internal consistency or test-retest reliability) and validity (convergent or discriminant validity, known-groups validity, or responsiveness data). These data will be summarized to capture the current state of the literature on measuring financial hardship in caregivers of individuals with ADRD. Data analysis is anticipated to be completed by November 2025.

## Discussion

ADRD is insidious and progressive, with a clinical course that is longer than many other degenerative conditions. Not surprisingly, the costs (financial and otherwise) of caring for an individual with ADRD are compounded and cumulative [27,28]. There also are many "hidden" costs associated with ADRD and ADRD-related care: costs to caregiver physical

and mental health [29–34]; lost income (for both the family members and the person living with ADRD) [14,29,30,35]; and impulsive spending or poor spending decisions (for the person living with ADRD) [21]. In addition, families contemplating supportive care placements for the person with ADRD (e.g., assisted living or nursing home care) must consider the economic impact on the family household. The lost income of the person with ADRD, as well as asset depletion from costs associated with assisted living can be financially devastating, especially for lower-income families.

While there are approaches to estimating the actual costs of ADRD (e.g., medical healthcare expenses), measures that assess "hidden" costs are lacking. Measures that do exist, such as measures of financial hardship that are not specific to people with a diagnosed health problem (including the Economic Strain Model [11], Financial Strain Scale [36], Financial Chronic Stress Scale [37], and COST-FACIT [12]), focus solely on individual- and/or household-level financial impact and do not capture intergenerational or cross-household financial support that may be exchanged in the case of ADRD. These measures are also not comprehensive; focusing only on single aspects of financial hardship, such as financial toxicity, but not other important aspects, such as financial stress or asset depletion. Current financial hardship measures are inappropriate for capturing the lived experience of people living with ADRD and their families, leaving a substantial gap in our ability to comprehensively characterize and quantify the impact of financial hardship in this context.

To address this need, the proposed study, which is informed by Lee and Cagle's [38] conceptual framework for understanding financial burden during serious illness, is designed to develop new measures to capture the objective (e.g., loss of job/income, resources, loss of savings/assets, care-related expenses, insurance and benefits, and budget constraints) and subjective (e.g., emotional impact) factors associated with financial hardship in caregivers for individuals with ADRD.

## Supporting information

**S1 Appendix. Example search.**
(DOCX)

**S2 Appendix. Proposed search criteria for AI-based extraction.**
(DOCX)

**S3 Table. Summary of features to be extracted from text.**
(DOCX)

## Author contributions

**Conceptualization:** Noelle E. Carlozzi, Christopher M. Graves, Jennifer A. Miner, Minal R. Patel.

**Data curation:** Jacqueline L. Freeman.

**Funding acquisition:** Noelle E. Carlozzi, Amanda N. Leggett, Jin-Shei Lai, Minal R. Patel.

**Investigation:** Noelle E. Carlozzi, Amanda N. Leggett, Jin-Shei Lai, Madison Fansher, Minal R. Patel.

**Methodology:** Noelle E. Carlozzi, Christopher M. Graves, Jacqueline L. Freeman, Jennifer A. Miner, Minal R. Patel.

**Project administration:** Christopher M. Graves, Jennifer A. Miner.

**Software:** Jacqueline L. Freeman.

**Supervision:** Noelle E. Carlozzi.

**Writing – original draft:** Noelle E. Carlozzi.

**Writing – review & editing:** Noelle E. Carlozzi, Christopher M. Graves, Jacqueline L. Freeman, Amanda N. Leggett, Jennifer A. Miner, Jin-Shei Lai, Madison Fansher, Minal R. Patel.

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
