## [Decision Letter · Decision Letter 0]

30 Jun 2025

Dear Dr. Carlozzi,

Thank you for submitting your manuscript to PLOS ONE. After careful consideration, we feel that it has merit but does not fully meet PLOS ONE’s publication criteria as it currently stands. Therefore, we invite you to submit a revised version of the manuscript that addresses the points raised during the review process.

We look forward to receiving your revised manuscript.

Kind regards,

Forgive Avorgbedor

Academic Editor

PLOS ONE

Journal Requirements:

“Work on this manuscript was supported by grant number R01AG088345(PI: Carlozzi, N.E.) from the National Institutes of Health (NIH), National Institute on Aging. Funders did not play a role in the development of the protocol. Dr. Leggett is also funded by NIAP30AB072931.”

Reviewers' comments:

Reviewer's Responses to Questions

**Comments to the Author**

1. Does the manuscript provide a valid rationale for the proposed study, with clearly identified and justified research questions?

Reviewer #1: Yes

2. Is the protocol technically sound and planned in a manner that will lead to a meaningful outcome and allow testing the stated hypotheses?

Reviewer #1: Yes

3. Is the methodology feasible and described in sufficient detail to allow the work to be replicable?

Reviewer #1: Yes

4. Have the authors described where all data underlying the findings will be made available when the study is complete?

Reviewer #1: Yes

5. Is the manuscript presented in an intelligible fashion and written in standard English?

Reviewer #1: Yes

You may also provide optional suggestions and comments to authors that they might find helpful in planning their study.

Reviewer #1: This protocol is well-structured and follows recognized standards (PRISMA-P/ScR), and the research question is relevant and timely. As this is a protocol paper, the methodological structure is largely fixed, and there is limited value in suggesting design changes at this stage.

However, as a reader, I would benefit from a clearer conceptual justification of why existing financial hardship instruments developed for cancer and other chronic illnesses are insufficient or inappropriate for use in Alzheimer’s disease and related dementias (ADRD). The authors briefly mention differences in household dynamics and duration of disease, but these important distinctions could be further developed.

Specifically, I encourage the authors to elaborate more explicitly on:

How intergenerational and cross-household financial support systems in ADRD differ from those in cancer care.

Why measures like the COST-FACIT or Economic Strain Model fail to capture critical aspects of financial hardship in ADRD caregiving.

What types of measurement domains might be uniquely required in the context of ADRD.

Expanding this rationale will not only strengthen the justification for the proposed new PRO system but also provide valuable guidance for future users or developers of ADRD-specific instruments.

**Do you want your identity to be public for this peer review?** For information about this choice, including consent withdrawal, please see our Privacy Policy

Reviewer #1: **Yes: ** keiichi abe

---

## [Author Response · Author response to Decision Letter 1]

28 Jul 2025

Editor’s Comments:

Editor Comment 1: Please ensure that your manuscript meets PLOS ONE's style requirements, including those for file naming

Response: We have confirmed that the submitted manuscript meets all specifications for style and formatting. We have named the resubmitted files as requested.

Editor Comment 2: Thank you for stating the following financial disclosure:

“Work on this manuscript was supported by grant number R01AG088345(PI: Carlozzi, N.E.) from the National Institutes of Health (NIH), National Institute on Aging. Funders did not play a role in the development of the protocol. Dr. Leggett is also funded by NIAP30AB072931.”

Response: We have added the above statement to the financial disclosure statement in the manuscript (lines 224-228) as well as the cover letter, confirming that the funders had no role in the study or manuscript.

Editor Comment 3: Please amend either the title on the online submission form (via Edit Submission) or the title in the manuscript so that they are identical.

Response: We have amended the title in the manuscript to match the title in the online submission form.

Editor Comment 4: Please review your reference list to ensure that it is complete and correct. If you have cited papers that have been retracted, please include the rationale for doing so in the manuscript text, or remove these references and replace them with relevant current references. Any changes to the reference list should be mentioned in the rebuttal letter that accompanies your revised manuscript. If you need to cite a retracted article, indicate the article’s retracted status in the References list and also include a citation and full reference for the retraction notice.

Response: We have reviewed the reference list, and confirm that it is complete and correct. No cited papers have been retracted, and the only changes to the reference list have been to add additional references in response to specific reviewer comments below.

Reviewer 1’s Comments:

Reviewer 1 Comment 1: This protocol is well-structured and follows recognized standards (PRISMA-P/ScR), and the research question is relevant and timely. As this is a protocol paper, the methodological structure is largely fixed, and there is limited value in suggesting design changes at this stage.

Response: Thank you for this feedback.

Reviewer 1 Comment 2: However, as a reader, I would benefit from a clearer conceptual justification of why existing financial hardship instruments developed for cancer and other chronic illnesses are insufficient or inappropriate for use in Alzheimer’s disease and related dementias (ADRD). The authors briefly mention differences in household dynamics and duration of disease, but these important distinctions could be further developed.

Specifically, I encourage the authors to elaborate more explicitly on:

How intergenerational and cross-household financial support systems in ADRD differ from those in cancer care.

Why measures like the COST-FACIT or Economic Strain Model fail to capture critical aspects of financial hardship in ADRD caregiving.

What types of measurement domains might be uniquely required in the context of ADRD.

Expanding this rationale will not only strengthen the justification for the proposed new PRO system but also provide valuable guidance for future users or developers of ADRD-specific instruments.

Response: Thank you for these suggestions. We have added additional references and text to the introduction to elaborate on these points (lines 69-95, 102-105).

---

## [Decision Letter · Decision Letter 1]

12 Aug 2025

Understanding financial hardship in families of people living with dementia: Protocol for a scoping review to identify subjective self-report measures that evaluate financial hardship

PONE-D-25-21569R1

Dear Dr. Carlozzi,

We’re pleased to inform you that your manuscript has been judged scientifically suitable for publication and will be formally accepted for publication once it meets all outstanding technical requirements.

Kind regards,

Forgive Avorgbedor

Academic Editor

PLOS ONE

Additional Editor Comments (optional):

Reviewers' comments:

Reviewer's Responses to Questions

**Comments to the Author**

1. Does the manuscript provide a valid rationale for the proposed study, with clearly identified and justified research questions?

Reviewer #1: Yes

2. Is the protocol technically sound and planned in a manner that will lead to a meaningful outcome and allow testing the stated hypotheses?

Reviewer #1: Yes

3. Is the methodology feasible and described in sufficient detail to allow the work to be replicable?

Reviewer #1: Yes

4. Have the authors described where all data underlying the findings will be made available when the study is complete?

Reviewer #1: Yes

5. Is the manuscript presented in an intelligible fashion and written in standard English?

Reviewer #1: Yes

You may also provide optional suggestions and comments to authors that they might find helpful in planning their study.

Reviewer #1: I have reviewed the revised introduction (lines 69–95, 102–105) and confirm that the authors have appropriately incorporated relevant literature and descriptions. The revisions adequately address the concerns raised in the previous review, and I find them acceptable.

**Do you want your identity to be public for this peer review?** For information about this choice, including consent withdrawal, please see our Privacy Policy

Reviewer #1: **Yes: ** Keiichi Abe

---

## [Editor Report · Acceptance letter]

PONE-D-25-21569R1

PLOS ONE

Dear Dr. Carlozzi,

I'm pleased to inform you that your manuscript has been deemed suitable for publication in PLOS ONE. Congratulations! Your manuscript is now being handed over to our production team.

Kind regards,

on behalf of

Dr. Forgive Avorgbedor

Academic Editor

PLOS ONE